# GPS: GENERAL PER–SAMPLE PROMPTER

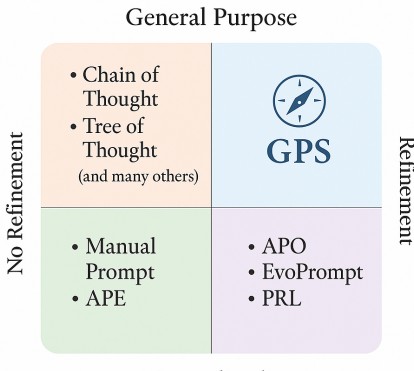 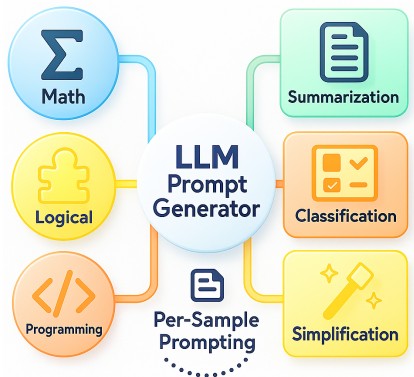

Figure 1: Left: Comparison of existing works to GPS. We propose the first automatic prompting method that is (i) general purpose, i.e. works without a task-specific training set and task-specific training and (ii) improves upon user-given prompts through refinement on a per-sample basis. Right: Overview of GPS, a general, per-sample prompter trained on mathematical, logical, and programming tasks. Once trained, it generates out-of-domain prompts for classification, summarization, and simplification. The model operates in a per-sample regime, producing a unique prompt for each input.

## ABSTRACT

LLMs are sensitive to prompting, with task performance often hinging on subtle, sometimes imperceptible variations in phrasing. As a result, crafting effective prompts manually remains challenging and time-consuming. Recent automatic prompting methods mitigate this difficulty but face three key limitations: (i) for each new task, they require large datasets to train good prompts; (ii) they rely on costly optimization loops that may take hours; (iii) they typically produce a single task-level prompt that does not adapt to the individual input problem to be solved.

We propose GPS, the first *general-purpose*, *per-sample* prompting method. Without any task-specific tuning, GPS generates a tailored prompt for each unseen input, improving performance across diverse tasks. The prompter is trained with reinforcement learning on a suite of training tasks and includes a novel regularization for effectively adapting to per-sample prompting. Finally, we employ Minimum Bayes Risk decoding to stabilize inference.

Empirically, GPS demonstrates competitive performance: we attain second best results among baselines on text simplification, third best results on summarization and on-par results on classification, while not training on any of these tasks, in contrast to the baselines. For in-domain prompting, we obtain sota on GSM8K. Our work shows the potential of a novel and effective paradigm for automatic prompting: generating adaptive, input-specific prompts without extensive optimization and without access to a task-specific training set. Code and data will be released upon acceptance.

# 1 INTRODUCTION

LLMs reveal their full potential only when guided by carefully designed prompts. Recent benchmarks show that model behaviour is highly sensitive to prompt phrasing Razavi et al. (2025), and this sensitivity persists even as model size increases Batorski et al. (2025); Guo et al. (2025).

Automatic prompt engineering has emerged to address this challenge. Early work searched for optimal prompts without iterative refinement Zhang et al. (2022), while later approaches introduced optimization loops based on evolutionary strategies or reinforcement feedback Pryzant et al. (2023); Guo et al. (2023); Batorski et al. (2025). Although effective, these methods must be re-run for every new task, making them impractical when prompts are needed on demand. In addition, each method requires the user to first construct a carefully curated dataset, often exceeding 1,500 samples, for the prompts to be usable, further limiting their practicality in real-world scenarios.

Moreover, a central shortcoming is task specificity: existing systems generate a single prompt per task, necessitating a new optimization cycle whenever the task changes. In practice, however, users expect effective prompts instantly. To address this, we introduce the first general prompter, a model trained to produce high-quality prompts for unseen tasks without requiring new datasets or costly optimization loops. To our knowledge, no prior work has provided a system working in this setting.

A last limitation concerns granularity. Most existing methods generate a single prompt for the entire task, implicitly optimizing for average-case performance. However, inputs within the same task can vary significantly in difficulty and characteristics, often benefiting from different demonstrations or instructions. As a result, a universal prompt may underperform on more challenging or atypical examples. This highlights the importance of moving beyond one-size-fits-all prompting toward more fine-grained, per-sample strategies.

For our approach we use Reinforcement Learning with Verifiable Rewards (RLVR) Lambert et al. (2024); Guo et al. (2025) and present GPS, a method that generates a separate prompt for every input instance of an unseen task. This fine-grained adaptation arises naturally within RLVR and requires no task-specific training.

However, incorporating observations into the prompt during reinforcement learning may lead the model to learn how to solve the task directly and embed the answer within the prompt itself. While this behavior is natural, it conflicts with the goal of building a general-purpose prompter and suppresses the model's ability to scale with model size. To address this, we propose a novel regularization mechanism that successfully encourages the prompt generator to produce prompts without embedding task-specific answers.

We conduct experiments in which GPS is trained on mathematical, logical reasoning, and coding tasks, and evaluated without any additional supervision on text classification, simplification, and summarization.

To summarize our contributions are as follows:

**Setting:** We present the first, to our knowledge, general-purpose prompter that generates high-quality prompts for unseen tasks without requiring any additional optimization or new training examples.

**Per-Sample:** Our method operates in a per-sample manner, generating a distinct prompt for each individual input.

**Regularization:** We introduce a novel regularization strategy that effectively prevents the model from embedding answers in the generated prompts. This encourages proper per-sample prompting and enables the approach to scale with the size of the model evaluator.

**Experiments:** We conduct extensive experiments demonstrating the effectiveness of our method on classification, summarization, simplification tasks as well as GSM8K.

**Ablations:** We provide an ablation study highlighting the importance of both per-sample prompting and the proposed regularization mechanism.

## 2 RELATED WORK

**Automated Prompt Engineering**   replaces manual prompt design with algorithmic generation to boost performance. APE (Zhou et al., 2022) generates prompt candidates from input–output pairs and selects the best, forgoing refinement once no further gains emerge. APO (Pryzant et al., 2023) refines prompts iteratively via natural-language critiques, using training-set examples and optimizing with minibatching, beam search, and bandit selection. EvoPrompt (Guo et al., 2023) applies evolutionary strategies to evolve prompts.RLPrompt (Deng et al., 2022) casts discrete prompt optimization as reinforcement learning: a lightweight policy network generates token-level prompts for a frozen LM and is trained via reward signals with stabilization techniques to handle noisy, delayed feedback. Batorski et al. (2025) introduces a RLVR approach to automatic prompt generation.

While effective, those methods above—performs task-specific optimization and must be run per dataset/task, whereas GPS aims to produce high-quality prompts for previously unseen tasks without task-level tuning.

**Per-sample Prompt Engineering**   A growing body of work investigates per-sample prompt generation to better tailor language models to individual inputs. *Instance-Wise Prompt Tuning* (IPT) (Jiang et al., 2022) learns input-dependent prompt embeddings, achieving fine-tuning-level performance with significantly fewer parameters. *Instance-Dependent Prompt Generation* (IDPG) (Wu et al., 2022) employs a lightweight generator to produce unique soft prompts for each input, surpassing fixed prompt tuning on a range of NLU tasks. Beyond NLP, per-sample prompting has also been explored in computer vision: *Domain-Adaptive Prompting* (DAP) (Jung et al., 2023) generates instance-level prompts at inference time to support rehearsal-free continual learning. Our method extends this paradigm by training per-sample prompts via reinforcement learning, a simple, general, and domain-agnostic approach applicable across diverse tasks.

## 3 METHOD

We train our model within a reinforcement learning paradigm. We adopt an architectural setup similar to Batorski et al. (2025) including a Prompt Generator and Evaluator Model, a similar reward formulation and optimization procedure.

Our approach consists of two LLMs:

- **Prompt Generator:** A trainable language model that generates prompts through a structured reasoning process (see Appendix . B).
- **Evaluator Model:** A frozen LLM that takes the generated prompt and produces a response.

Subsequently, we present a detailed breakdown of each component.

**Reward Function**   Our total reward is the sum of several sub-rewards: $r_{\text{token}}, r_{\text{structure}}, r_{\text{format}}$, and $r_{\text{alignment}}$.

*Token-level formatting reward:*

- Each correctly placed marker: `<think>`, `</think>`, `<answer>`, and `</answer>` earns a reward of $\frac{r_{\text{token}}}{4}$, provided if appears exactly once.
- If all four markers are used correctly, the generator receives the total token reward of $r_{\text{token}}$.
- A structural bonus $r_{\text{structure}}$ is awarded when the output exactly follows the format: `<think>` reasoning `</think>` `<answer>` final answer `</answer>`, ensuring a clean two-phase response.

*Evaluator Model rewards:*

- $r_{\text{format}}$: Adherence to a required pattern (e.g. multiple-choice options). Depends on the specific task.
- $r_{\text{alignment}}$: Measures task accuracy or other performance metrics, such as correctness, ROUGE, or SARI, depending on the task.

These components combine to form the overall reward:

$$R = r_{\text{token}} + r_{\text{structure}} + r_{\text{format}} + r_{\text{alignment}}. \tag{1}$$

**Prompt Generator** The generator $\pi_\theta^{\text{generator}}$ is conditioned on a base prompt and observation (See Fig. 4), and produces reasoning traces followed by candidate prompts. At each training iteration, the generator samples outputs $o_1, \ldots, o_n$, from which prompts $p_1, \ldots, p_n$ are extracted and evaluated by the Evaluator Model.

**Evaluator Model** The evaluator, denoted as $\pi^{\text{eval}}$, is a frozen LLM. It is queried using prompts generated by the Prompt Generator.

**Regularization** When the Prompt Generator receives both the prompt and the corresponding observation as input, there is a risk that it will learn to directly output the correct instead of letting the evaluation model answer. Such behavior defeats the goal of learning a transferable prompting strategy and limits effectiveness. The example of the prompt leakage is shown in Appendix G.

To mitigate this, we introduce two complementary regularization strategies:

- **LLM-based Regularization (Judge).** We employ an auxiliary frozen language model, referred to as the *Judge*, to verify whether a generated prompt contains the answer. If the Judge detects that the prompt leaks the solution, a penalty of $-1$ is applied to the reward. This discourages the Prompt Generator from embedding answers directly in prompts. The prompt template used for the Judge is shown in Appendix H.

- **Sample-Based Regularization.** After the Prompt Generator produces a prompt for a specific observation $x$, we evaluate that prompt with probability $p$ not on $x$ itself, but on a randomly selected subset $\{x_1, \ldots, x_n\}$ from the same task. This regularization encourages the generation of prompts that generalize beyond the original input. Increasing $p$ enhances robustness, though it may reduce the prompt's specificity to individual examples.

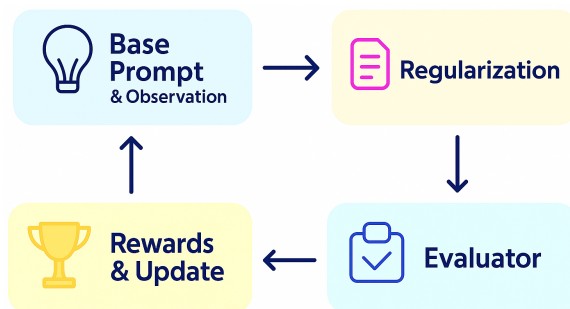

Figure 2: Training cycle of GPS. First, the *Prompt Generator* produces an initial prompt based on the given observation. This prompt is then regularized using either *Judge Regularization* or *Sample Regularization* to prevent label leakage, i.e., the inclusion of the correct answer within the prompt itself. The *Evaluator* then assesses the quality of the regularized prompt by measuring its accuracy and provides a reward signal. Finally, the model is updated based on this feedback to improve prompt quality over time.

These strategies strike a balance between leveraging information from specific samples and avoiding overfitting or solution leakage. During inference, the Prompt Generator creates prompts using only the specific example, without any regularization. The pseudocode for the training loop, as well as for the Judge and Sample Regularization components, is provided in Appendix A.

**Optimization** We update the Prompt Generator using Group Relative Policy Optimization (GRPO) (Shao et al., 2024), a variant of PPO that eliminates the need for a separate value network by computing baseline-adjusted, group-wise rewards. This offers memory efficiency and leverages the relative ranking of multiple prompt samples. Pseudo-code of the training loop is shown in Fig. 2.

**Decoding.** We view RLVR as a distribution shifter: training nudges the generator toward regions of the prompt space that yield useful outputs, but at inference time an unseen task induces a large hypothesis space, so committing to a single high-variance prompt is brittle. Therefore, we employ Minimum Bayes Risk (MBR) decoding Kumar & Byrne (2004) expressed purely in terms of a task-specific *utility* function.

For an input $x$, the Prompt Generator samples $N$ candidate prompts $c_1, \ldots, c_N$. The Evaluator maps each $(x, c_j)$ to an output $y_j$. Let $\mathcal{H}(x) = \{y_1, \ldots, y_N\}$. Given a utility $u : \mathcal{H}(x) \times \mathcal{H}(x) \to [0, 1]$, MBR selects

$$\hat{y} = \arg \max_{y \in \mathcal{H}(x)} \mathbb{E}_{y' \sim p(\cdot|x)}\big[u(y, y')\big]. \tag{2}$$

Since $p(\cdot \mid x)$ is unknown, we use the empirical distribution over the $N$ evaluator outputs. The empirical expected utility of $y_j$ is

$$\hat{U}(y_j) = \frac{1}{N} \sum_{k=1}^{N} u(y_j, y_k), \qquad \hat{y} = \arg \max_{y_j \in \mathcal{H}(x)} \hat{U}(y_j).$$

We consider two regimes:

- *Classification.* Use the agreement utility $u(y, y') = \mathbf{1}[y = y']$. Under the uniform empirical distribution, MBR reduces to majority voting:

$$\hat{y} = \arg \max_{y_j} \sum_{k=1}^{N} \mathbf{1}[y_j = y_k].$$

- *Generation (summarization/simplification).* Use a *reference-free consensus* utility
$$u(y, y') = \tfrac{1}{3}(\text{ROUGE-1}(y, y') + \text{ROUGE-2}(y, y') + \text{ROUGE-}L(y, y')) \in [0, 1],$$
and select
$$j^\star = \arg \max_j \ \frac{1}{N-1} \sum_{k \neq j} u(y_j, y_k), \qquad \hat{y} = y_{j^\star}.$$

All quantities are computed *per input* $x$ using only the $N$ evaluator outputs for that $x$; no ground-truth labels, reference texts, or statistics from other test examples are used.

## 4 EXPERIMENTS

All experiments are conducted on two NVIDIA A100 GPUs (40 GB each). Each model is trained for 48 hours, following the setup of Batorski et al. (2025). We use the Qwen2.5-7B-Instruct model (Yang et al., 2024) as both the Prompt Generator and the Evaluator for GPS as well as for all baseline benchmarks, ensuring a fair comparison.

Our models are fine-tuned using Group Relative Policy Optimization (GRPO) with parameters $\epsilon = 0.2$, $\beta = 0.04$, and a weight decay of 0.1. Additionally, we apply Low-Rank Adaptation (LoRA) (Hu et al., 2022) with a learning rate of $5 \times 10^{-5}$, setting the scaling factor $\alpha = 32$ and the rank $r = 8$.

**Training Datasets**  We train GPS on three diverse tasks: mathematical reasoning, logical reasoning, and programming motivated by evidence that task diversity and difficulty enhance model reasoning Muennighoff et al. (2025). For each task, we define simple rule-based *alignment* and *format* rewards.

- **Mathematical Reasoning:** Alignment = 1 if the answer matches ground truth; Format = 1 if the output follows expected syntax (e.g., integer, Yes/No).
  - **GSM8K** Cobbe et al. (2021): 8.5K grade school math word problems requiring multi-step reasoning.
  - **DeepMath-103K** He et al. (2025): a corpus of formally verified mathematics problems covering diverse topics; we train on a randomly chosen subset of 3,000 examples whose answers are either integers or binary (`Yes`/`No`).
- **Logical Reasoning:** Alignment = 1 if the selected answer is correct; Format = 1 if the response follows the required multiple-choice format.
  - **CommonSenseQA** Talmor et al. (2019): 12K commonsense QA questions based on Concept-Net.
  - **OpenBookQA** Mihaylov et al. (2018): 6K science QA questions requiring world knowledge.
  - **MedQA** Jin et al. (2020): Real-world medical board exam questions covering clinical and biomedical topics.
- **Programming:** Alignment = 2 if the generated function passes all unit tests and uses the correct name; Format = 0 (not enforced).
  - **MBPP** Austin et al. (2021): 1K entry-level Python problems with reference solutions and test cases.

**Methods**  We compare GPS against both human-written, task-specific prompts and a variety of general-purpose prompt engineering methods. It is important to highlight that the benchmark methods described below are tuned separately for each dataset, whereas GPS is a unified prompting approach trained without access to any examples from the classification, summarization, or simplification datasets. For a fair comparison, we use the same model Qwen2.5-7B Instruct ((Yang et al. (2024))), as the prompt generator and evaluator across all baseline methods.

- **MI (Manual Instruction)** (Zhang et al., 2022): Handcrafted prompts written by humans, aiming to boost performance on individual tasks using manually designed instructions.
- **NI (Natural Instruction)** (Mishra et al., 2021): Similar to MI, crowd-sourced human-written instructions.
- **APE (Automatic Prompt Engineer)** (Zhou et al., 2022): APE automatically generates a set of instruction candidates using an LLM, and then selects the most effective prompt based on its downstream performance with a target model. Candidate prompts are not refined during the optimization process.
- **APO (Automatic Prompt Optimization)** (Pryzant et al., 2023): APO uses an iterative feedback loop using beam search to refine prompts without relying on gradients, treating prompt tuning as a black-box optimization problem.
- **EvoPrompt** (Guo et al., 2023): Uses evolutionary strategies: selection, crossover, and mutation to evolve a pool of discrete prompts, discovering high-performing prompts.
  - **DE (Differential Evolution)**: Uses differential evolution to traverse the prompt search space.
  - **GA (Genetic Algorithm)**: Applies classic genetic operators such as selection, crossover, and mutation to cultivate progressively better prompts.
- **PRL (Prompts from Reinforcement Learning)** (Batorski et al., 2025): PRL applies a reinforcement learning loop to automatically generate and optimize prompts.

**Summarization**  We evaluate GPS on an abstractive summarization task, where the model must extract and condense the most salient information from a given dialogue. The goal is to produce concise summaries that retain essential content while filtering out irrelevant or redundant details.

Experiments are conducted on the SAMSum dataset (Gliwa et al., 2019), a curated corpus of English chat dialogues resembling real-world messenger conversations. These dialogues, created by linguists to reflect informal and natural exchanges, are paired with manually written abstractive summaries.

Table 1: Text summarization results averaged over three runs.

| Method | ROUGE-1 | ROUGE-2 | ROUGE-L |
|---|---|---|---|
| MI | 32.76 | 10.39 | 28.97 |
| APE | $37.12_{\pm2.02}$ | $12.97_{\pm0.74}$ | $33.32_{\pm1.68}$ |
| GA | $39.69_{\pm1.76}$ | $14.47_{\pm1.00}$ | $35.84_{\pm1.63}$ |
| DE | $33.91_{\pm4.04}$ | $12.53_{\pm1.47}$ | $31.05_{\pm3.79}$ |
| PRL | $42.47_{\pm0.83}$ | $16.17_{\pm0.24}$ | $37.73_{\pm0.36}$ |
| GPS-J | $38.08_{\pm0.74}$ | $13.07_{\pm0.44}$ | $34.09_{\pm0.61}$ |
| GPS-SR-0.1 | $40.03_{\pm0.11}$ | $14.36_{\pm0.13}$ | $35.91_{\pm0.19}$ |

To measure summarization performance, we employ the standard ROUGE metrics (Lin, 2004): **ROUGE-1** measures unigram overlap, assessing content selection, **ROUGE-2** bigram overlap, evaluating coherence and phrasing and **ROUGE-L** the longest common subsequence, reflecting structural and fluency alignment.

The results in Table 1 show that both GPS–J and GPS–SR–0.1 perform well on summarization. GPS–J surpasses APE, and GPS–SR–0.1 ranks among the top three methods overall. We argue that in summarization we see quite strongly the benefit of per-sample prompting, our prompt can adapt to the topic, style etc. of the text to be summarized in contrast to non-sample specific methods.

Table 2: Results on task simplification averaged over three runs.

| Method | SARI |
|---|---|
| MI | 43.77 |
| APE | $45.33_{\pm0.83}$ |
| GA | $46.25_{\pm0.47}$ |
| DE | $45.79_{\pm0.35}$ |
| PRL | $52.26_{\pm3.51}$ |
| GPS-J | $52.09_{\pm0.22}$ |
| GPS-SR-0.1 | $48.10_{\pm0.66}$ |

**Simplification**  We evaluate GPS on sentence simplification using the ASSET dataset (Alva-Manchego et al., 2020), a crowdsourced corpus curated for testing rewriting capabilities such as lexical paraphrasing, sentence splitting, deletion, and reordering. Each original sentence is paired with multiple human-written simplifications, offering diverse reference outputs that enable comprehensive evaluation of model performance.

To measure the quality of simplification, we use the SARI metric (Xu et al., 2016). SARI compares model output to the reference simplifications and also to the original sentence, scoring the additions, deletions, and preserved elements in the output. It aligns well with human assessments of simplicity, making it a trusted and effective metric for this task.

The results of the sentence simplification task are shown in Table 2. GPS–J attains the second-highest average SARI score across all methods, while GPS–SR–0.1 ranks third. Remarkably, despite being trained exclusively on out-of-domain tasks such as mathematics, logic, and programming, GPS outperforms several in-domain baselines tuned for simplification such as APE, APO, and Evo-Prompt. An illustrative example of a generated simplification prompt is provided in Appendix E. Overall, these results highlight the strong out-of-domain generalization ability of GPS and demonstrate the effectiveness of per-sample prompt generation even without task-specific supervision.

**Classification** We evaluate the performance of GPS on a diverse set of language understanding classification tasks, including the following:

- **Binary sentiment classification**: SST-2 (Socher et al., 2013), MR (Pang & Lee, 2005), and CR (Hu & Liu, 2004) datasets for identifying whether a sentence conveys a `positive` or `negative` sentiment.
- **Multiclass sentiment classification**: SST-5 (Socher et al., 2013) extends binary sentiment classification to five sentiment levels: `terrible`, `bad`, `okay`, `good`, or `great`. This gives a finer granularity in sentiment detection compared.
- **Question classification**: The TREC dataset (Voorhees & Tice, 2000) requires determining the semantic category of a question, choosing from six options: `Description`, `Entity`, `Expression`, `Human`, `Location`, or `Number`.
- **News topic classification**: AG's News (Zhang et al., 2015) consists of news headlines and descriptions categorized into four domains: `World`, `Sports`, `Business`, and `Tech`.
- **Subjectivity analysis**: The SUBJ dataset (Pang & Lee, 2004) involves labeling sentences as either `subjective` or `objective`, for distinguish personal opinions from factual statements.

Results are presented in Table 3. Overall, GPS-J performs better GPS-SR-0.1, and GPS-J occasionally surpasses methods that are specifically tailored for individual tasks. In particular, we achieve a top-3 position on SST-5 and top-2 and top-3 positions on AG News. These findings suggest that GPS is especially effective on text classification tasks with a larger number of classes. In such cases, the boundaries between classes are less distinct, which allows per-prompt sampling to more effectively enhance performance. On the other hand, GPS performs poorly on Subj (64–65%), where fine-grained, dataset-specific cues and fragile decision boundaries make distinguishing subjective from objective statements especially challenging. Without access to in-domain exemplars, per-sample prompting alone is insufficient to fully capture these nuances.

Table 3: Accuracy (%) on seven text-classification benchmarks, averaged over three runs. For each dataset, the best, second-best, and third-best scores are highlighted in red, orange, and yellow, respectively. Standard deviations ($\pm$) are shown in script-style next to each mean.

| Method | SST-2 | CR | MR | SST-5 | AG News | TREC | Subj | Avg |
|---|---|---|---|---|---|---|---|---|
| MI | 92.70 | 87.25 | 87.40 | 52.31 | 82.29 | 69.20 | 57.95 | 75.59 |
| NI | 95.77 | 91.50 | 90.85 | 51.90 | 83.43 | 66.60 | 68.10 | 78.31 |
| APO | $93.71_{\pm0.25}$ | $93.48_{\pm0.24}$ | $89.97_{\pm1.37}$ | $53.94_{\pm0.29}$ | $83.73_{\pm0.31}$ | $71.30_{\pm1.90}$ | $69.80_{\pm5.96}$ | 79.42 |
| APE | $91.23_{\pm0.66}$ | $92.87_{\pm0.02}$ | $89.90_{\pm0.94}$ | $49.37_{\pm5.66}$ | $82.58_{\pm1.20}$ | $77.07_{\pm1.61}$ | $73.92_{\pm1.39}$ | 79.56 |
| GA | $94.65_{\pm1.04}$ | $92.75_{\pm0.40}$ | $90.45_{\pm0.72}$ | $53.76_{\pm1.13}$ | $82.24_{\pm1.00}$ | $79.20_{\pm2.83}$ | $74.93_{\pm3.12}$ | 81.14 |
| DE | $93.29_{\pm0.34}$ | $93.38_{\pm0.19}$ | $89.98_{\pm0.24}$ | $55.25_{\pm0.37}$ | $82.18_{\pm1.04}$ | $76.47_{\pm0.38}$ | $73.08_{\pm4.95}$ | 80.52 |
| PRL | $96.32_{\pm0.04}$ | $92.83_{\pm0.24}$ | $91.27_{\pm0.05}$ | $56.21_{\pm0.15}$ | $84.36_{\pm0.08}$ | $77.07_{\pm2.36}$ | $76.90_{\pm0.95}$ | 82.14 |
| GPS-J | $94.25_{\pm1.20}$ | $90.65_{\pm0.05}$ | $89.15_{\pm0.38}$ | $55.16_{\pm0.36}$ | $84.04_{\pm0.02}$ | $72.80_{\pm0.60}$ | $64.20_{\pm2.25}$ | 78.61 |
| GPS-SR-0.1 | $92.98_{\pm0.19}$ | $90.50_{\pm0.38}$ | $88.70_{\pm0.05}$ | $55.14_{\pm1.13}$ | $84.21_{\pm0.34}$ | $68.20_{\pm0.20}$ | $65.10_{\pm0.28}$ | 77.83 |

**GSM8K** In this experiment, we present the performance of GPS on the GSM8K dataset. This benchmark is particularly interesting as it evaluates how GPS acquires reasoning abilities, given that its training primarily involves reasoning tasks. It is important to note that GSM8K samples were included in the training set; however, we evaluate our method on the held-out test set, which was not

seen during training. The results, summarized in Table 4, show that GPS-SR-0.1 achieves the highest accuracy on GSM8K, while GPS-J ranks among the top three models. These findings suggest that GPS not only learns to solve GSM8K problems included in the training set, but also benefits from transfer effects derived from training on other reasoning tasks. This highlights the effectiveness of GPS in enhancing performance on reasoning benchmarks.

**Ablation Study: Effect of Regularization on DeepMath Reasoning Tasks**  In this experiment, we investigate the role of regularization in improving generalization on the challenging DeepMath benchmark, which contains mathematically sophisticated problems requiring precise multi-step reasoning. We hypothesize that without regularization, the Prompt Generator may overfit to the training setup by embedding answers directly into the prompts thereby diminishing the benefits of using larger evaluator models. We compare the performance of GPS using two regularization strategies: Judge Regularization and Sample Regularization (with probability 0.1) against two baselines:

Table 4: GSM8K Results.

| Method | Acc. |
|---|---|
| MI | 78.20 |
| APE | $83.43_{\pm1.98}$ |
| GA | $81.62_{\pm1.38}$ |
| DE | $79.52_{\pm0.45}$ |
| PRL | $86.15_{\pm0.55}$ |
| GPS-J | $84.45_{\pm0.93}$ |
| GPS-SR-0.1 | $87.55_{\pm0.42}$ |

- **No Reg.**: A variant trained without any regularization. This model is free to insert answers directly into the prompt, leading to potential leakage.

- **Base**: A static, handcrafted prompt used uniformly across all benchmarks, without any learned refinement.

To assess the generalization capacity of these methods, we evaluate the prompts generated by each model using increasingly capable evaluators: Qwen2.5-{7B|32B|72B}-Instruct. We sample 2000 previously unseen DeepMath tasks and evaluate model accuracy under each configuration. The results, presented in Figure 3, demonstrate: (i) GPS outperforms the base prompt, (ii) As the evaluator becomes larger, prompt leakage hurts more and more, while our regularizations produce generalizable prompts that scale better and benefit more from more capable evaluators. We

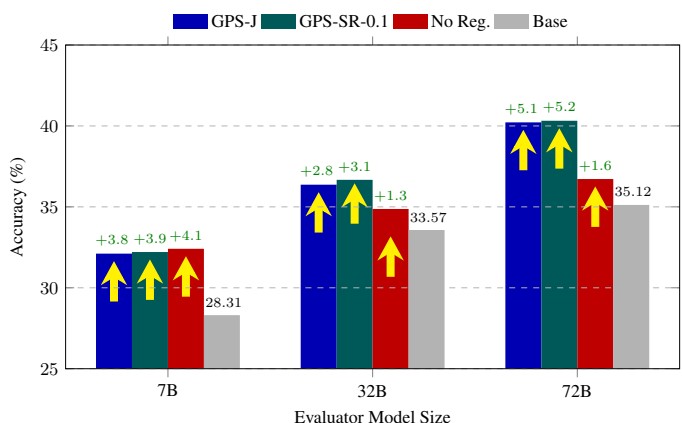

Figure 3: Comparison of accuracy on the DeepMath benchmark across different regularization strategies and evaluator sizes.

have found that some prompts generated by GPS are already precise and well-formed, yet the 7B evaluator lacks the capacity to solve the corresponding task. This suggests that the limitation lies not in the prompt itself, but in the evaluator's reasoning ability. An illustrative example of such a case where a prompt generated by GPS–SR–0.1 is incorrectly answered by the 7B evaluator but correctly solved by the 72B evaluator is presented in Appendix I.

**Ablation: Effect of Sample Regularization Probability**  We study the effect of varying the swap probability in our sample regularization mechanism. A high probability may cause the model to disregard the observation entirely when generating prompts, while a very low probability increases the risk of solution leakage, where the prompt implicitly encodes the answer. In Table 5 we report results on swap probabilities of 0.1, 0.2 and 0.5 and also compare against Judge regularization on subjectivity classification, summarization and simplification. While there is no clear best setting, 0.1 and Judge regularization are overall good values.

**Ablation: Effect of Minimum Bayes Risk Decoding**  We evaluate GPS–SR–0.1 both with and without MBR across the subjectivity, summarization, and simplification tasks. The results, sum-

Table 5: Performance of regularization strategies and per–sample prompting across three NLP tasks. SR = Sample Regularization; J = LLM-Judge regularization; J+SR-0.2 = Judge + Sample Regularization with swap probability 0.2; No-PSP = no per–sample prompting; No-MBR = decoding without Minimum Bayes Risk; Llama = model trained with the Llama-3.1-8B-Instruct backbone.

| Task | Metric | Method | | | | | | | |
|---|---|---|---|---|---|---|---|---|---|
| | | SR-0.1 | SR-0.2 | SR-0.5 | J | J+SR-0.2 | No-PSP | NO MBR | Llama |
| Subj | Accuracy | $65.10_{\pm 0.28}$ | $62.55_{\pm 1.31}$ | $66.00_{\pm 2.25}$ | $64.20_{\pm 2.25}$ | $62.95_{\pm 2.35}$ | $70.40_{\pm 1.47}$ | $64.62_{\pm 0.33}$ | $60.90_{\pm 0.7}$ |
| Simplification | SARI | $48.10_{\pm 0.66}$ | $46.42_{\pm 1.74}$ | $48.84_{\pm 0.50}$ | $52.09_{\pm 0.22}$ | $49.33_{\pm 0.01}$ | $44.11_{\pm 1.93}$ | $47.25_{\pm 1.11}$ | $52.04_{\pm 0.34}$ |
| Summarization | ROUGE-1 | $40.03_{\pm 0.11}$ | $40.12_{\pm 0.58}$ | $39.85_{\pm 1.99}$ | $38.08_{\pm 0.74}$ | $38.26_{\pm 0.61}$ | $36.88_{\pm 0.78}$ | $38.00_{\pm 0.20}$ | $40.46_{\pm 0.15}$ |
| | ROUGE-2 | $14.36_{\pm 0.13}$ | $13.89_{\pm 0.43}$ | $14.10_{\pm 1.30}$ | $13.07_{\pm 0.44}$ | $13.43_{\pm 0.07}$ | $12.39_{\pm 0.62}$ | $12.97_{\pm 1.26}$ | $14.62_{\pm 0.17}$ |
| | ROUGE-L | $35.91_{\pm 0.19}$ | $35.47_{\pm 0.55}$ | $35.54_{\pm 1.56}$ | $34.09_{\pm 0.61}$ | $34.28_{\pm 0.55}$ | $33.11_{\pm 1.78}$ | $33.66_{\pm 1.6}$ | $35.91_{\pm 0.23}$ |

marized in Table 5, show that while the model already achieves competitive performance without MBR, applying MBR consistently yields further improvements across all tasks.

**Ablation: Effect of Per-Sample Prompting**  We train the model on the same datasets as GPS, but with a key difference: during training, we only show the base prompt, i.e. the prompt to be enhanced without including any accompanying observations. As a result, the model has access to a limited number of observations, since the dataset now consists of only a few base prompts. Nevertheless, it still receives rich reward signals derived from those observations. During training, we craft a single prompt for the currently processed task and sample 10 observations from that specific task on which the prompt is evaluated. We then compare models trained with and without per-sample prompting across three tasks: simplification, summarization, and subjectivity classification. Results in Table 5 indicate that removing per-sample prompting attains the strongest performance on subjectivity classification, while remaining competitive—on par with APE for summarization and with a manual prompt for simplification. This pattern suggests that, for classification tasks with relatively coarse decision boundaries, a single well-tuned prompt can be sufficient. By contrast, for generation tasks such as summarization and simplification, which require finer control, GPS with PSP delivers consistent improvements.

**Ablation: Cross-model performance**  We assess the cross-model generality of GPS by replacing the Qwen training backbone with LLAMA-3.1-8B-INSTRUCT (AI@Meta, 2024), motivated by reports that even with random rewards Qwen can exhibit notable gains on certain tasks (Shao et al., 2025). To isolate backbone effects, we keep all training settings fixed and apply the same Sample Regularization (probability 0.1). We train and evaluate on summarization, simplification, and subjectivity classification.

Results are shown in Table 5. With the exception of subjectivity classification, training with Llama surpasses training with Qwen across tasks. These findings indicate that our pipeline is sensible and that GPS does not derive its improvements from Qwen specific oddity or the spurious reward phenomenon, but rather transfers effectively across model backbones.

## 5 CONCLUSIONS & LIMITATIONS

We have shown the viability of general-purpose zero-shot per-sample prompting, reaching competitive results on text summarization, simplification and classification and GSM8K, while being trained exclusively on mathematical, logical and programming tasks. We argue that such a setting is more realistic, since in practice we do not have access to a large training set with ground truth answers. We also hope to stimulate development of automatic prompting methods for this regime.

In order to close the gap to automatic prompting methods that use task-specific optimization, we estimate that fast training-free synthesizing of few-shot examples might be helpful. Another avenue is better regularization: While ours was effective in suppressing prompt leakage, this was achieved at the cost of also inhibiting to some extent adaptation to the current sample. An avenue for further research is advanced regularization and other mechanisms for this problem.

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

## A  GPS Pseudo Codes

In this section we provide:

1. **Algorithm 1** – overall training loop of GPS
2. **Algorithm 2** – Judge regularization
3. **Algorithm 3** – Sample regularization

---

**Algorithm 1** GPS Training

---

**Require:**
    Prompt generator $\pi_\theta^{\text{gen}}$
    Frozen evaluator $\pi^{\text{eval}}$
    Dataset $\mathcal{D}$
    Iterations $T$
    Prompts per step $n$
**Ensure:** trained parameters $\theta$
1: **for** $i \leftarrow 1$ **to** $T$ **do**
2:     $(x, b) \leftarrow \text{SAMPLE}(\mathcal{D})$
3:     $o_{1:n} \leftarrow \pi_\theta^{\text{gen}}(b, x)$
4:     $p_{1:n} \leftarrow \text{EXTRACTPROMPTS}(o_{1:n})$
5:     $r_{1:n} \leftarrow$
        $\text{COMPUTEREGULARIZEDREWARDS}\,(p_{1:n}, x, \pi^{\text{eval}})$
6:     $\theta \leftarrow \text{GRPOUPDATE}(\theta, r_{1:n})$
**return** $\theta$

---

**Algorithm 2** JUDGEREGULARIZATION

---

**Require:**
    Observation $x$
    Base prompt $b$
    Prompt generator $\pi_\theta^{\text{gen}}$
    Frozen evaluator $\pi^{\text{eval}}$
    Frozen judge $\pi^{\text{judge}}$
1: $p \leftarrow \pi_\theta^{\text{gen}}(x, b)$         ▷ candidate prompt
2: $R \leftarrow \pi^{\text{eval}}(x, p)$
3: $q \leftarrow \text{WRAPWITHTEMPLATE}(p)$     ▷ template from App. H
4: $y \leftarrow \pi^{\text{judge}}(q)$     ▷ `"1"` if leak, else `"0"`
5: **if** $y = 1$ **then**
6:     **return** $R - 1$     ▷ apply leakage penalty
7: **else**
8:     **return** $R$

---

## B  System and user prompt

In this section we provide the system and user prompt used in GPS models.

## C  Base Prompts

In this section we provide all base prompts that are used for each task type in GPS.

**OpenBookQA**

Choose one of the correct answers. Return only the correct response ['A', 'B', 'C', 'D'] without any additional text.

**CommonSense & MedQA**

Choose one of the correct answers. Return only the correct response ['A', 'B', 'C', 'D'] without any additional text.

**Algorithm 3** SAMPLEREGULARIZATION

**Require:**
    Observation $x$ of task $t$
    Base prompt $b$ for $t$
    Training set $\mathcal{D}$
    Swap probability $p_{\text{swap}}$
    Prompt generator $\pi_\theta^{\text{gen}}$
    Frozen evaluator $\pi^{\text{eval}}$
1:  $p \leftarrow \pi_\theta^{\text{gen}}(x, b)$
2:  Draw $u \sim \mathcal{U}(0, 1)$
3:  **if** $u < p_{\text{swap}}$ **then**                       ▷ swap branch
4:     $\hat{\mathcal{D}}_t \leftarrow \text{SAMPLE}(\mathcal{D})$
5:     $R \leftarrow 0$
6:     **for all** $x_j \in \hat{\mathcal{D}}_t$ **do**
7:         $R \leftarrow R + \pi^{\text{eval}}(x_j, p)$
8:     **return** $R$
9:  **else**                                  ▷ no swap
10:     **return** $\pi^{\text{eval}}(x, p)$

---

**System prompt**

A conversation between User and Assistant. The user asks a question, and the assistant solves it. The assistant first thinks about the reasoning process in the mind and then provides the user with the answer. The reasoning process and answer are enclosed within
`<think>` ... `</think>` `<answer>` ... `</answer>`.

**User Prompt**

Your task is to refine a base prompt for another model that performs a math task. You will be given the base prompt and the observation for which the prompt should be enhanced. Improve the instructions to enhance the model's performance. Return only the enhanced prompt.
BASE PROMPT: Solve this riddle and return ONLY the integer answer:
OBSERVATION: Natalia sold clips to 48 of her friends in April, and then she sold half as many clips in May. How many clips did Natalia sell altogether in April and May?

Figure 4: System prompt (top) and user prompt (bottom) used in the prompt refinement task. The system prompt defines the expected format of responses, while the user prompt instructs the assistant to refine a base prompt for improved performance on a specific observation.

**GSM8K**

Solve this riddle and return ONLY the integer answer.

**DeepMath**

Solve this riddle and return ONLY the integer answer or 'Yes', 'No' without any other text.

**MBPP**

Solve this coding task. Provide the python code that solves this problem (with return statements). Return this function and nothing else. Do not provide any usage examples. Every argument should be defined inside the function.

**Summarization**

How would you rephrase that in a few words?

**Simplification**

Simplify the text.

**SST-2**

Please perform Sentiment Classification task. Given the sentence, assign a sentiment label from ['negative', 'positive']. Return label only without any other text.

**Simplification**

Simplify the text.

**CR & MR & SST–2**

Please perform Sentiment Classification task. Given the sentence, assign a sentiment label from ['negative', 'positive']. Return label only without any other text.

**SST-5**

Please perform Sentiment Classification task. Given the sentence, assign a sentiment label from ['terrible', 'bad', 'okay', 'good', 'great']. Return label only without any other text.

**AG's News**

Please perform News Classification task. Given the news item, assign a label from ['World', 'Sports', 'Business', 'Tech']. Return label only without any other text.

**TREC**

Please perform Question Classification task. Given the question, assign a label from ['Description', 'Entity', 'Expression', 'Human', 'Location', 'Number']. Return label only without any other text.

**SUBJ**

Please perform Subjectivity Classification task. Given the sentence, assign a label from ['subjective', 'objective']. Return label only without any other text.

## D  EFFECTIVE PROMPT FOR SST-5 AND AG'S NEWS

In this section, we present well-crafted example prompts for the SST-5 sentiment classification task and AG's News topic classification task.

### SST-5 PROMPTS

Below are example prompts for the SST-5 sentiment classification task.

**Observation**

Take care of my cat offers a refreshingly different slice of Asian cinema.

**Prompt**

Please perform a Sentiment Classification task. For each sentence, assign a sentiment label from the following list: ['terrible', 'bad', 'okay', 'good', 'great']. The label should be determined based on the overall tone and content of the sentence. Focus on identifying whether the sentiment is positive, negative, or neutral. Return the label only without any additional text or explanations.
For example:
- "The movie was a complete disaster." should be classified as 'terrible'.
- "It was a mediocre experience." should be classified as 'okay'.
- "This film exceeded all my expectations!" should be classified as 'great'.

### AG'S NEWS PROMPT

Below is an example prompt for the AG's News topic classification task.

**Observation**

FT.com – Shares in Sohu.com, a leading US-listed Chinese internet portal, fell more than 10 percent on Friday after China's biggest mobile-phone network operator imposed a one-year suspension on its multimedia messaging services because of customer spam complaints.

**Prompt**

Please perform the News Classification task.
Choose exactly one label from: ['World', 'Sports', 'Business', 'Tech'].

**Guidelines**
- Look for the dominant theme (global affairs, athletics, finance/economics, or technology).
- If a story mixes topics, prefer the one most central to the headline and body.
- Return **only** the label, nothing else.

Example (for illustration):
Headline: "LeBron James leads Lakers to victory in NBA opener."
Correct output → Sports

## E   PROMPTS FOR SIMPLIFICATION

In this section, we provide example prompt used for the sentence simplification task.

**Observation**

The Kindle 2 features 16-level grayscale display, improved battery life, 20 percent faster page-refreshing, a text-to-speech option to read the text aloud, and overall thickness reduced from 0.8 to 0.36 inches (9.1 millimeters).

**Prompt**

Simplify the following description of the Kindle 2 for a general audience. Focus on the most important features and present them in a straightforward manner, avoiding technical terms and details that might confuse the reader.
Key features to highlight include:
- Display type and quality
- Battery life improvement
- Reading speed
- Text-to-speech capability
- Size reduction

Please provide a simplified version that covers these points:

## F   PROMPTS FOR SUMMARIZATION

In this section, we provide example prompt used for the sentence summarization task.

**Observation**

Mr. Williams: Ms. Blair, would you like to go for a coffee?
Ms. Blair: I thought you'd never ask.
Mr. Williams: That's outstanding. Do you have a favourite coffee place?
Ms. Blair: I actually do. It's near the square in a side alley.
Mr. Williams: I think I know which one. Let's say 2 p.m.?
Ms. Blair: Sounds great. See you there :)

**Prompt**

Summarize the brief conversation between Mr. Williams and Ms. Blair in a concise manner, capturing the main points and preserving the conversational tone. Include the following details:

1. Mr. Williams invites Ms. Blair for coffee.
2. Ms. Blair expresses enthusiasm for the invitation.
3. They discuss meeting at a specific coffee place.
4. They agree on a time and confirm the meeting.

## G   PROMPTS WITHOUT REGULARIZATION

In this section we provide exmaple of prompts that leakage i.e. in the prompts there is already an answer.

**Observation**

not since japanese filmmaker akira kurosawa's ran have the savagery of combat and the specter of death been visualized with such operatic grandeur .

**Leakage Prompt (No Regularization)**

"Please perform a Subjectivity Classification task. Given the sentence, determine if it is subjective or objective and assign a label from ['subjective', 'objective']. Return the label only without any additional text or explanation.
Example:
Input: not since japanese filmmaker akira kurosawa's ran have the savagery of combat and the specter of death been visualized with such operatic grandeur.
Expected Output: subjective

**Prompt after Regularization**

Please perform a subjectivity classification task on the given sentence. Determine if the sentence expresses opinions, feelings, or is more factual and straightforward. Assign the label 'subjective' if the sentence contains personal views, emotions, or interpretations. Assign the label 'objective' if the sentence contains facts or is neutral without personal bias. Return only the label without any additional text.

**Ground Truth**

subjective

Figure 5: Comparison between a flawed and a regularized prompt setup for subjectivity classification. The observation is the actual user input. The leakage prompt embeds the correct answer within an example that mirrors the test input, effectively leaking the label into the prompt. This kind of leakage compromises evaluation integrity, as it allows the model to extract or memorize the answer without performing the task. The regularized prompt, on the other hand, avoids including the target label and better reflects a fair testing setup. The ground truth shows the expected model output. Regularization techniques are essential for mitigating this type of leakage and ensuring reliable performance evaluation.

## H  PROMPT FOR JUDGE LLM

In this section, we provide the prompt for the Judge LLM, which is responsible for regularization.

**Judge Prompt**

You will receive a single text input: a "candidate prompt" that is intended for another LLM to solve a specific problem. Your job is to decide whether this candidate prompt already contains the solution (i.e., it reveals the answer) to that problem.
If the candidate prompt does include already the solution, reply with exactly: 1
If the candidate prompt does not include the solution (it only contains instructions, context, examples, or background without giving the answer), reply with exactly: 0
Return only '1' or '0' without any additional text.
Examples:
Example 1: No solution embedded Candidate prompt: "Summarize the following paragraph in one sentence: 'Artificial intelligence (AI) refers to machines that can perform tasks typically requiring human intelligence, such as recognizing speech, translating languages, and making decisions.' Provide a concise summary without revealing the original paragraph verbatim." $\rightarrow$ 0
Example 2: Solution embedded Candidate prompt: "Translate the English sentence 'The cat sits on the mat' into French. The answer is 'Le chat est assis sur le tapis'. After that, explain each word's role in the sentence." $\rightarrow$ 1
The prompt to evaluate:

## I  INTERESTING FAILURES DESPITE GOOD PROMPTS

In this section, we present notable failure cases where the generated prompts are well-structured, logically sound, and provide a clear step-by-step solution to the task. Despite this, the evaluator still selects an incorrect answer, highlighting limitations in the downstream model's reasoning or alignment.

**Question**

Find the longest geometric progression with a common ratio greater than 1 that can be formed from the set $\{100, 101, 102, \ldots, 1000\}$. What is the length of this progression?

**Prompt**

Solve the riddle step-by-step:
1. Understand the set range: $\{100, 101, 102, \ldots, 1000\}$.
2. Identify the common ratio greater than 1 and ensure the geometric progression doesn't exceed the maximum of the set. Here, the common ratio needs to be larger and the numbers should be successive members of a geometric progression.
3. Iterate through the set and detect the longest geometric progression. This requires a careful examination of numbers and their relationships.
The highest integer answer is found through these steps to be:
Solve this riddle and return ONLY the integer answer or 'Yes', 'No' without any other text.

**Prediction**

No

**Ground Truth**

6

## J   USE OF LLMS

LLMs were used exclusively to improve the clarity and readability of the texts.

