# OpenReview forum: "GPS: General Per–Sample Prompter"
_ICLR.cc/2026/Conference — Submitted to ICLR 2026_

### Official Review · Reviewer_3EL1 · 2025-10-30

**Soundness:** 3
**Presentation:** 3
**Contribution:** 2
**Rating:** 4
**Confidence:** 3

**Summary:**

General-per-sample prompter is a trained model for generating task- and sample- specific prompts for novel samples and tasks. This improves on prior optimization approaches by dynamically generating sample-specific prompts and generalizing beyond in-distribution training tasks. The model is trained using RL with a novel regularizer intended to prevent reward hacking that might result from the prompter providing the answer directly to the LLM. Experimental results are presented showing the prompter's capability to generalize across tasks, and ablations are performed on the effects of varying regularization, decoding, and per-sample prompting methods.

**Strengths:**

GPS adapts to the task and generates a unique prompt for each sample.
GPS aims to generalize to unseen tasks, making it more versatile.
A novel regularization approach prevents reward hacking, which could generalize to other RL settings.
The prompter is decoupled from the evaluator, enabling a smaller model to boost the performance of a large model.

**Weaknesses:**

In general the experimental results show that the method is competitive but PRL seems to be a stronger method overall- the authors should discuss the relative advantages of their approach compared to PRL. Was PRL evaluated on unseen tasks?
Training tasks are largely reasoning tasks while test tasks are mainly classical NLP tasks, underscoring the need for ablations to understand whether GPS generalizes well to more complex tasks. The GSM8K results are encouraging but the absence of a holdout reasoning task is a liability.

**Questions:**

How did you choose the tasks you trained on? Did you perform any cross-validation by mixing the training tasks?

Explain the distinction from PRL and the benefits? Is PRL trained directly on the test tasks, explaining the superior performance?

Is it necessary to calibrate r_{alignment} so that task rewards are balanced?

Have you performed any ablation over N in the decoding phase?  Compare with a baseline best-of-N prompting scheme?

It took me some time to figure out that GPS-J and GPS-SR-0.1 correspond to the two regularization methods- be sure to indicate this clearly in section 3.

---

> ### Author Response · Authors · 2025-11-20
>
> We thank the reviewer for his insightful review. Below we address the raised concerns:
>
> - *W1 "method is competitive but PRL seems to be a stronger method overall"*
>
>   We agree that PRL is overall stronger in our experiments, and this is expected: PRL is trained separately on each evaluation task, whereas GPS is a single, generic prompter that has not seen these tasks during training. As a result, GPS operates in a strictly harder setting. We therefore do not claim that GPS should outperform task-specific methods; rather, we view it as encouraging that GPS can approach the performance of PRL and other task-specific prompting approaches without any task-specific training, and even match or surpass them on some benchmarks.
>
> - *W2 "PRL evaluated on unseen tasks"*
>
>   By design, PRL is a task-specific method: for each downstream task it requires a dedicated training set and an RL optimization loop on that task. In this sense, PRL is *not* evaluated on “unseen” tasks in our regime, because it cannot be applied without first training on task-specific data. In contrast, GPS is trained once on a separate set of meta-tasks and then applied zero-shot to new tasks for which no training data are available.
>
> - *W3: "Training tasks are largely reasoning tasks while test tasks are mainly classical NLP tasks, need for ablations"*
>
>   This design choice is intentional: a key goal of our work is to test cross-domain robustness, i.e., whether a prompter trained on verifiable reasoning tasks (math/logic/code) can transfer to classical NLP tasks (summarization, simplification, classification) that never appear during training. To further probe this question, in the rebuttal we have also evaluated GPS on additional benchmarks beyond those in the main paper, including both in-domain reasoning tasks (DeepMath, MedQA) and out-of-domain ones (MATH500). The new results, summarized in the General Answer section "1. Comparison with GRACE and additional benchmarks", show that GPS remains competitive in both regimes, providing additional evidence that the learned prompting policy generalizes across domains rather than overfitting to a narrow set of training tasks.
>
>
> - *Q1 "How did you choose the tasks you trained on? Did you perform any cross-validation by mixing the training tasks?"*
>
>   We appreciate the reviewer’s question. Our training mixture was chosen to consist of mathematical, logical-reasoning, and coding tasks that admit verifiable, programmatic rewards (exact numeric answers, multiple-choice correctness, unit tests), which are crucial for stable RL/RLVR training and are empirically known to transfer well to broader reasoning benchmarks (see general response, Sec. 2: “Choice of mathematical, coding and commonsense reasoning tasks”). We did not perform an exhaustive cross-validation over all possible task combinations, instead, we fixed this mixture a priori based on these considerations. Exploring alternative and larger meta-task collections, including systematic mixing of training tasks, is an interesting extension that we plan to investigate in future work.
>
> - *Q2 "Explain the distinction from PRL and the benefits? Is PRL trained directly on the test tasks, explaining the superior performance?"*
>
>   Yes, PRL is trained directly on each evaluation task: we follow the EvoPrompt setup and use its train/validation/test splits to optimize a separate PRL policy per task. This task-specific training regime largely explains why PRL often attains higher scores.
>
>   In contrast, GPS is trained once on a disjoint set of meta-tasks (math/logic/code) and then applied zero-shot to all evaluation tasks, without access to their training data and without any further tuning. The comparison therefore contrasts two different regimes:
>   - PRL: per-task RL with task-specific supervision, optimized directly on the evaluation distribution.
>   - GPS: a single, general-purpose per-sample prompter, reused across tasks with no task-specific data.
>
>   The benefit of GPS is precisely this reusability and data-free deployment: once trained, it can be applied to new tasks “out of the box,” whereas PRL must be retrained whenever a new task is introduced.

---

> ### Author Response · Authors · 2025-11-20
>
> - *Q3 "Is it necessary to calibrate r_{alignment} so that task rewards are balanced?"*
>
>   We sample tasks uniformly during training. For each task, the alignment reward $r_{\text{alignment}}$ is given by the corresponding task metric (e.g., accuracy, exact match), which we normalize to lie in $[0, 1].$ or in $[0,2]$ depending on the task (whether we consider $r_{\text{format}}$)
>
>   This normalization keeps alignment rewards on a comparable scale across tasks, so no additional per-task calibration is needed.
>
> - *Q4 "Have you performed any ablation over N in the decoding phase? Compare with a baseline best-of-N prompting scheme?"*
>
>   In the current submission we study two endpoints of this spectrum:
>   - the **No-MBR** setting in Table 5, which corresponds to standard single-sample decoding (`N = 1`) under Sample Regularization, and
>   - our **GPS-SR-0.1** setting, where we draw `N = 15` samples and apply MBR over the resulting evaluator outputs.
>
>   The gap between these two columns therefore already quantifies the benefit of moving from `N = 1` to a moderate multi-sample regime.
>
>   For classification tasks, our MBR with agreement utility is already very close in spirit to a best-of-`N` scheme, as it effectively selects the label that is most consistently supported across the `N` samples. For generation tasks, however, naive best-of-`N` tends to favor high-variance outliers, whereas our consensus-based MBR is designed to be more robust.
>
> - *Q5 "It took me some time to figure out that GPS-J and GPS-SR-0.1 correspond to the two regularization methods- be sure to indicate this clearly in section 3."*
>
>   Thank you for pointing this out. In a revised version, we will explicitly state in Section 3 that GPS-J and GPS-SR-0.1 denote the variants with Judge Regularization and Sample Regularization (with swap probability 0.1), respectively.

---

> > ### Comment · Reviewer_3EL1 · 2025-11-27
> >
> > Thank you for your detailed rebuttal. I now understand better the distinction from PRL and its limitations.  I have bumped my overall score and the contribution score.

---

> > > ### Author Response · Authors · 2025-11-28
> > >
> > > We thank the reviewer for engaging with our rebuttal and for raising the score. If there are any remaining concerns we could address that might justify a further increase, we would be very happy to do so.

---

### Official Review · Reviewer_TVfW · 2025-10-31

**Soundness:** 3
**Presentation:** 2
**Contribution:** 2
**Rating:** 4
**Confidence:** 4

**Summary:**

The paper proposes GPS (General Per-Sample Prompter), a reinforcement-learning–trained prompt generator that, at inference, produces a unique prompt per input for an unseen task. Two regularization schemes (an LLM “Judge” that penalizes answer leakage and a sample-swap scheme) aim to prevent the generator from embedding solutions in the prompt. Inference uses Minimum Bayes Risk (MBR) selection over multiple candidate prompts/evaluator outputs. Trained on math/logic/programming tasks, GPS is evaluated out-of-domain on summarization (SAMSum), simplification (ASSET), and classification (SST-2/-5, MR, CR, AG News, TREC, SUBJ). Results show 2nd–3rd best performance on summarization/simplification and competitive classification averages, plus strong in-domain GSM8K, with ablations for regularization, MBR, per-sample prompting, backbone (Qwen vs Llama), and evaluator scale.

**Strengths:**

- The problem setup and the paper is well-motivated.
- Method simplicity and plausibility: two-LLM setup (generator/evaluator), GRPO optimization, and explicit reward decomposition (format/structure/accuracy).
- Broad evaluation and ablations: out-of-domain generalization (trained on reasoning; tested on summarization/simplification/classification), backbone swap (Qwen→Llama), effect of regularization probability, per-sample vs no-per-sample, and MBR on/off.

**Weaknesses:**

- Effectiveness:
   - Underperforms PRL on most tasks. The central claim is that a task-agnostic, per-sample prompter should be at least competitive with task-specific prompt RL (PRL). Yet across the main tables (summarization, simplification, multi-task classification), GPS typically lags PRL. This weakens the paper’s core effectiveness claim.
   - Ask for a controlled, cost-normalized head-to-head. Please report a training budget comparison against PRL, including GPU hours, latency and token cost.

- Novelty / Positioning

   - “First general-purpose per-sample prompter” is not well established. Instance-level prompt optimization is not new, and per-sample demonstration selection (retrieval-augmented ICL) is widely studied. The paper should more clearly delineate how GPS differs from (i) instance-dependent prompt tuning/selection and (ii) retrieval-based ICL that adapts the context per input.

   - Some other works generate per-instance keywords such as "https://arxiv.org/abs/2302.11520" is also considerred as per-sample PO.

- Missing Baselines:

  - Include some demonstration-selection baselines, since they are also per-sample strategies.

- Additional Evidence that would Strengthen the Paper:

   - What do optimized prompts look like?

   - Judge validation: How reliable is your Judge?

**Questions:**

See above.

---

> ### Author Response · Authors · 2025-11-20
>
> We thank the reviewer for the constructive criticism. Below, we address each raised issue.
>
> - *W1: "Effectiveness"*
>   - *"Underperforms PRL on most tasks"*
>
>     While GPS often underperforms PRL, this is expected: PRL is trained separately on each evaluation task, whereas GPS has never seen data from these tasks during training. We therefore do not anticipate a task-agnostic, cross-task prompter to consistently outperform task-specific methods, even when it adapts at the per-sample level. Instead, we view it as encouraging that GPS achieves performance close to PRL and other task-specific prompters, and even surpasses some of these baselines on certain benchmarks.
>
>   - *"Cost-normalized comparison"*
>
>     A strictly cost-normalized comparison is difficult, because PRL and GPS make different assumptions about how often a method is reused. PRL is trained separately for each new task (up to 48 hours per task in our setup), whereas GPS is pre-trained once (also for 48 hours) and can then be applied to any new task without further training. For a new task, PRL thus incurs an additional training cost, while GPS can be used out of the box, which makes GPS more economical when amortized over many tasks.
>     At inference time, the trade-off is reversed: PRL produces a single task-level prompt that can be cached and used with a single forward pass of the evaluator, while GPS generates a per-sample prompt, requiring two LLM calls per input (one to the generator, one to the evaluator) and therefore higher per-sample latency. This overhead is inherent to per-sample prompting. In terms of token cost, GPS tends to generate shorter prompts on average than GRACE/APO/PRL, which often include in-context examples. The table below reports average prompt lengths (in tokens) for GPS on our benchmarks:
>
> Summarization | Simplification | CR| MR | AG News | SST-2 | SST-5 | Subj | TREC |
> | --- | ----- | ----- | ----- | ----- | ----- | ----- | ----- | ----- |
> 33.54 | 35.65 | 51.80 | 52.88 | 57.51 | 52.35 | 64.13 | 55.32 | 55.52 |
>
> - *W2: "Novelty/Positioning: First general-purpose per-sample prompter” is not well established."*
>
>   We appreciate the reviewer’s comments and agree that both instance-level prompt optimization and per-sample demonstration selection have been actively studied. We will soften and clarify our “first general-purpose per-sample prompter” claim accordingly. Concretely, prior work such as IDPG and related instance-dependent prompt tuning methods learn per-instance soft prompts or input-conditioned prompt vectors, but do so separately for each downstream task, using that task’s labeled data and gradient access to the underlying model.
>
>   Retrieval-augmented in-context learning, on the other hand, adapts the context per input via example retrieval from a task-specific datastore, but still assumes a corpus of task-specific examples and focuses on selecting demonstrations rather than learning a cross-task prompting policy.
>
>   Directional Stimulus Prompting (DSP) is indeed very close in spirit: it trains a small policy LM to generate discrete, instance-specific “directional stimulus” tokens for each downstream task, using supervised examples and RL signals from that task. In other words, applying DSP to a new task requires collecting task-specific data and retraining or fine-tuning the policy LM.
>
>   In contrast, GPS is designed as a single, general-purpose per-sample prompter that is (i) trained once on a mixture of verifiable reasoning/meta tasks (math/logic/code), (ii) then applied zero-shot to a broad set of unseen tasks (summarization, simplification, classification) without any labeled data or retriever corpus for those evaluation tasks, and (iii) interacts with the evaluator purely through natural-language prompts in a black-box setting. We therefore do not claim that GPS is the first method to perform per-instance prompting in any sense; rather, our contribution is to show that a task-agnostic, RL-trained per-sample prompter can be meta-trained on verifiable reasoning tasks and still transfer competitively to heterogeneous out-of-domain NLP tasks, without per-task tuning, soft-prompt access, or task-specific retrieval. We will revise the text to make this narrower notion of “general-purpose per-sample prompter” explicit and to more clearly situate GPS relative to DSP, IDPG/MetaPrompter, and retrieval-based ICL.

---

> ### Author Response · Authors · 2025-11-20
>
> - *W3: "Missing baselines, demonstration selection"*
>
>   We agree that per-sample demonstration selection (retrieval-based in-context learning) is a closely related line of work and a natural comparison point. However, these methods operate under a different assumption: they require access, at test time, to a pool of input–output pairs from the same downstream task in order to retrieve demonstrations and insert them into the prompt.
>
>   In contrast, the regime we focus on in this paper is strictly no-task-data: when a new task arrives, GPS only receives the base instruction and the current input $x$, and it never sees any labeled or unlabeled examples from that task. This is also the reason why our baselines (PRL, GRACE, EvoPrompt, APO, etc.) are all task-level prompt optimizers that assume a training split for each evaluation task, while GPS is trained once on meta-tasks and then used zero-shot.
>
>   Because of this mismatch in assumptions, a direct empirical comparison to demonstration-selection methods would be inherently asymmetric: to run those methods we would have to give them extra supervision (a task-specific demonstration pool) that GPS is explicitly **not** allowed to use in our setting. Conceptually, demonstration selection is therefore complementary rather than competing with our method.
>
>   We will make this distinction explicit in the revised version by (i) clearly stating in Section 1 that our primary focus is the no-task-data regime, and (ii) adding a short discussion of retrieval-based demonstration selection in the related work as a stronger baseline when task-specific exemplars are available.
>
> - *W4: "Additional evidence"*
>
>   - "What do optimized prompts look like?"**
>
>     We agree that it is important to make the behaviour of GPS as transparent as possible. Concrete examples of optimized prompts are already included in the appendix: Appendix D (SST-5 and AG’s News), Appendix E (simplification), Appendix F (summarization), and Appendix G (subjectivity). In addition, Appendix I presents failure cases where the generated prompts are well-structured but the evaluator still answers incorrectly, illustrating both the strengths and limitations of the learned prompts.
>
>   - "How reliable is your Judge?"
>
>     We thank the reviewer for raising this point. We have manually inspected a substantial number of prompts produced under Judge regularization and did not observe any instances of label leakage according to our assessment. Qualitatively, Appendix G provides a representative before/after example where Judge regularization removes an implicit answer from the prompt while preserving useful task instructions.
>
>     Quantitatively, Figure 3 in the main paper provides indirect but strong evidence for the effectiveness of Judge regularization. If the Judge systematically failed, the prompt generator would still be able to encode answers in the prompt. In that case, increasing the evaluator size would yield only limited gains, since the answer would already be fed into the prompt. Instead, we observe that models trained with Judge (and Sample) regularization benefit significantly more from larger evaluators than the unregularized variant. This behaviour is consistent with Judge regularization successfully suppressing leakage and forcing the evaluator to do the actual reasoning rather than simply reading off an answer from the prompt.

---

### Official Review · Reviewer_hbtD · 2025-11-01

**Soundness:** 2
**Presentation:** 2
**Contribution:** 2
**Rating:** 2
**Confidence:** 4

**Summary:**

This paper proposes a new framework called GPS, to customize the prompt for the inputs to the large language models (LLMs). The customization focuses on the instance-level and is done by a general prompt generator without finetuning on any data from the targeted downstream tasks. More specifically, a general prompt generator is trained via GRPO on the prompt generation task with the data from mathematical, logical, and programming tasks. Two regularization methods are developed to discourage the generator from inserting the answer to the input into the prompt template and hack the rewards. Experimental results on several NLP tasks show that the proposed method can achieve comparable performance with the existing prompt customization methods finetuned on the downstream tasks.

**Strengths:**

The strengths of the paper are listed as follows.
1. The paper is well motivated. The prompt design can affect the performance of the LLM and a general-purpose generator can reduce the human effort in prompt design significantly.
2. The two regularization methods are reasonable and reflect deep insights from the authors in the prompt customization task.
3. It is great to see that the authors provided a detailed ablation analysis in their methods, providing more insights into their method.

**Weaknesses:**

The weaknesses of the paper are listed as follows.
1. The experimental results are not convincing enough. I illustrated this point as follows:
a. The evaluated downstream tasks are relatively naïve. More complicated tasks should be evaluated to show the effectiveness of GPS such as commonsense reasoning, information extraction and long-context understanding.
b. The performance of in-domain performance in the math, logical reasoning and coding is also important to understand whether GPS is underfitting to the meta-tasks.
c. It is not clear why Table 1 and Table 2 did not include the baseline named NI for the comparison. Besides, according to Table 3, the training-based GPS does not show a clear advantage over the NI, a baseline that designs the prompt by humans.
2. The writing of the paper can be improved to make it easier to follow. For example, the contribution can be summarized into three key points, focusing on technical novelty, methodology, and evaluation. Secondly, since the training pipeline is based on PRL, the authors should highlight the difference between GPS and PRL in the related work section. Thirdly, in the method section, it would be better if the authors could formulate the prompt generation problem as an optimization problem to clearly define the objective, constraints, the learnable parameters, input and output to the problem.
3. According to the proposed training pipeline, the general-purpose prompt generator is not model-agnostic and thus limits its impact. For example, the generated prompt may be overfit to the model family of the generator. So when the target LLM is not in the same family as the generator, the generated prompts may be suboptimal. The experiments do not provide enough evidence to address this concern since the authors did not compare with the existing baselines in their ablation study on cross-model performance.
4. Some important technical details are not clear. I raised the questions in the Questions section of the review.

**Questions:**

My questions are listed as follows.
1. What is the average prompt length GPS generated for each evaluated task? Will it generate a much longer prompt than the baselines and incur much higher latency for online inference?
2. What is the latency for a single sample inference with GPS? Will the introduction of an LLM-based prompt generator cause heavy overhead on the latency?
3. Could you provide more explanation about why picking math, logical reasoning and coding tasks as the meta-tasks to train the GPS? Why not the other tasks? Is the performance of GPS sensitive to the choice of meta-tasks and the relevant training dataset?
4. What are the differences between GPS and IPT or IDPG mentioned in the related work? Why not compare GPS with them in the experiments?
5. For equation (1), is there any scaling factor for each reward term?
6. In the proposed GRPO training, how to define a group when calculating the group average advantage?
7. In the proposed GRPO training, what is the decoding strategy of prompt generator and evaluator?

---

> ### Author Response · Authors · 2025-11-20
>
> We thank you for the deep analysis of our paper and answer the raised issues as follows:
>
> - *W1: "Experimental results are not convincing enough"*
>   - *a. "Evaluated downstream tasks are relatively naive"*
>   - *b. "The performance of in-domain performance is also important"*
>
>     Our experiments follow commonly used benchmarks in the prompt generation literature. In particular, GSM8K (Table 4) and DeepMath (Figure 3) both require multi-step, non-trivial reasoning rather than simple classification. For this rebuttal, we have additionally included comparisons on DeepMath and MedQA with all baselines, thereby adding two more in-domain tasks and showing that GPS is not underfitting the meta-training setting. We further report results on MATH500, a challenging out-of-domain mathematical reasoning benchmark. Together, these experiments provide evidence that GPS is competitive on both in-domain and out-of-domain on higher-difficulty tasks; please see 1. Comparison with GRACE and additional benchmarks in the General answers section.
>
>     We would welcome concrete suggestions for additional benchmarks that the reviewers consider particularly important for a camera-ready version.
>
>
>   - *c. "It is not clear why Table 1 and Table 2 did not include the baseline named NI for the comparison. Besides, according to Table 3, the training-based GPS does not show a clear advantage over the NI, a baseline that designs the prompt by humans."*
>
>     For summarization and simplification, NI prompts are simply not available in the experimental setup we follow. In these tables we adhere exactly to the evaluation protocol of EvoPrompt, which also does not include NI for SAMSum (summarization), ASSET (simplification), or any task other than classification. Introducing new, hand-crafted NI-style prompts for these tasks would require us to design additional human baselines ourselves, making the comparison less controlled and no longer directly comparable to prior work.
>
>     For the classification results (Table 3), GPS-J indeed only slightly outperforms NI, which we see as evidence that carefully engineered human prompts can be very strong baselines. Our goal is not to argue that a general, training-based method must always beat a well-designed human prompt on every dataset, but rather that GPS can match such prompts while requiring no task-specific prompt design. Moreover, our ablations (Table 5) show that per-sample prompting is not universally beneficial: for SUBJ in particular, a single task-level prompt (the “No-PSP” variant) performs best, and none of the per-sample methods—including GPS—improves over it. This supports the view that instance-level prompting is highly effective on some tasks (e.g., GSM8K, SAMSum, ASSET), but not uniformly superior across all settings.

---

> ### Author Response · Authors · 2025-11-20
>
> - *W2: "Writing can be improved, e.g. for contributions, difference to PRL and formulation of objective"*
>
>   Thank you for this suggestion. We agree that the exposition can be clarified in these three places, and we will revise the paper accordingly.
>
>   *Contributions.* In a revised version, we will restructure the introduction to explicitly list the main contributions along three axes:
>   (i) the setting (a general-purpose, per-sample prompter that does not require any training data for the downstream task),
>   (ii) the method (an RLVR-based training pipeline for per-sample prompt generation with two regularization mechanisms to avoid leakage), and
>   (iii) the evaluation (systematic out-of-domain experiments on summarization, simplification, and classification, plus in-domain GSM8K and new results on DeepMath, MedQA, and MATH500).
>
>   *Difference to PRL.* We will also make the distinction to PRL much more explicit in the related work and experimental sections. Both PRL and GPS use RL-based optimization, but they operate in fundamentally different regimes:
>
>   PRL trains a separate prompt policy for each downstream task, using a task-specific training set and reward, and is therefore tuned directly on the evaluation distribution.
>
>   GPS, in contrast, is pre-trained once on a suite of meta-tasks (math, logic, programming) and then frozen. At test time, GPS receives only the base prompt and the input sample for a new task and does not see any training data from the downstream task. All benchmarks like classification, summarization, or simplification are thus genuinely unseen for GPS, whereas PRL has been explicitly optimized for each of them.
>
>   *Formulation of objective.* We will add a concise formal definition of the learning objective. Formally, we model each training task $t \in \mathcal{T}$ as a triple $(\mathcal{D}_t, b_t, R_t)$, where $\mathcal{D}_t$ is a data distribution over inputs $x$, $b_t$ is a base prompt, and $R_t$ is a task specific reward function. The prompt generator takes $b_t$ and $x$ and produces a prompt $p$:
>
> $$p \sim \pi_\theta^{\text{gen}}(\cdot \mid b_t, x).$$
>
>   A frozen evaluator then uses $p$ and $x$ to produce an output $y$:
> $$y \sim \pi^{\text{eval}}(\cdot \mid p, x).$$
>
>   The pair $(p, y)$ is scored by the task reward
>   $R_t(x, p, y) = R(t, x, p, y)$. Our goal is to choose $\theta$ to maximize
>   the average reward over tasks and inputs:
>
>   $$\text{max}_\theta \ \text{E} [R(t, x, p, y) ].$$
>
>   where we sample $t \in \mathcal{T}$, $x \sim \mathcal{D}_t$, $p \sim \pi^{gen}$ and $y \sim \pi^{\text{eval}}$.
>
>   We decompose the reward into simple components:
>   $R(t, x, p, y) = r_{\text{token}}(p) + r_{\text{structure}}(p) + r_{\text{format}}(t, y) + r_{\text{alignment}}(t, x, y),$
>
>   where $r_{\text{token}}$ and $r_{\text{structure}}$ enforce formatting of the generator output, and $r_{\text{format}}$ and   $r_{\text{alignment}}$ measure task-specific formatting and correctness. We optimize $\theta$ with GRPO while keeping $\pi^{\text{eval}}$ fixed.

---

> ### Author Response · Authors · 2025-11-20
>
> - *W3: "Prompt generator is not model-agnostic and thus limits its impact"*
>
>   We agree that model-agnostic behavior is desirable. To test this, we explicitly retrained GPS with prompt generator as Llama-3.1-8B-Instruct instead of Qwen2.5-7B-Instruct (see the last column of Table 5), keeping the rest of the setup unchanged. In this setting, GPS continues to work well and, on summarization and simplification, the Llama-based variant even slightly outperforms the Qwen-based one. This indicates that our training procedure is not tied to a single backbone and that GPS can be transferred to a different model family by swapping the underlying LLM.
>
> - *Q1: "Average prompt length"*
>
>   Average prompt lengths for each task are reported in the table below. The prompts generated by GPS are not noticeably longer than manually designed instructions and are substantially shorter than prompts that include in-context examples (e.g., GRACE, PRL or APO). This shows that GPS does not rely on very long prompts to obtain its gains.
>
> Summarization | Simplification | CR| MR | AG News | SST-2 | SST-5 | Subj | TREC |
> | --- | ----- | ----- | ----- | ----- | ----- | ----- | ----- | ----- |
> 33.54 | 35.65 | 51.80 | 52.88 | 57.51 | 52.35 | 64.13 | 55.32 | 55.52 |
>
>
> - *Q2 "Latency"*
>
>   This is an important point. There is an inherent trade-off between per-sample prompting and per-task prompting (our baselines). For GPS, each query requires two LLM calls: one to the prompt generator and one to the evaluator, so latency is higher than for methods that use a single, pre-computed task-level prompt.
>
>   However, this overhead is unavoidable for any per-sample method: if the prompt depends on the specific input, it must be computed at inference time. In our setting, the additional latency corresponds exactly to one extra LLM call per request, which we view as the natural cost of gaining per-sample adaptivity.
>
> - *Q3: "Task selection for training"*
>
>   We thank the reviewer for this question. We refer to the general section "2. Choice of mathematical, coding and commonsense reasoning tasks."

---

> ### Author Response · Authors · 2025-11-20
>
> - *Q4: "Difference between IPT/IDPG and GPS"*
>
>   We thank the reviewer for raising this. We will clarify the relationship in the final version.
>
>   1. **What is being trained and where the parameters live**
>
>    - IPT (Instance-wise Prompt Tuning) and IDPG are parameter-efficient fine-tuning methods for the *target model itself*:
>
>      - IPT learns instance-wise soft prompt embeddings that are prepended to the input of a frozen pretrained language model, while all original model weights remain fixed.
>
>      - IDPG adds a lightweight prompt generator module $G$ on top of a frozen encoder (e.g., RoBERTa-style backbones) and trains $G$ per task to produce instance-dependent soft prompts for each input sentence.
>
>    - GPS, in contrast, trains a separate, general-purpose generator LLM with RLVR. The evaluator / target LLM is never updated. GPS only interacts with the evaluator via natural-language text prompts (black-box setting), not via additional soft-prompt parameters.
>
>   2. **Prompt representation: soft embeddings vs natural language**
>
>    - IPT/IDPG: prompts are **continuous embeddings**; they are not directly human-readable and require architectural access to the model’s embedding / transformer stack.
>
>    - GPS: produces **discrete, natural-language prompts**. This is compatible with API-only models and keeps prompts interpretable and transferable across different evaluator backbones.
>
>   3. **Data assumptions and training regime**
>
>    - IPT is trained with **task-specific supervised data** for each downstream task and optimized via gradient descent on a prompt-tuning loss, with the base model frozen.
>
>    - IDPG similarly assumes supervised training data \(D = \{(x_i, y_i)\}_{i=1}^n\) for each downstream task, and the prompt generator is optimized with standard supervised objectives on that task.
>
>    - GPS is trained **once** on a mixture of verifiable reasoning meta-tasks (math/logic/coding) with RLVR, and at test time sees **no labeled data** from the evaluation tasks (summarization, simplification, classification). It is explicitly a zero-shot, cross-task meta-prompter.
>
>   We did not include IPT or IDPG in our experiments because they operate under assumptions that differ substantially from the setting we study, which would make a direct comparison non-trivial and, in our view, somewhat unfair. Both IPT and IDPG assume supervised training data and gradient access for **each downstream task**: they are designed as parameter-efficient ways to adapt a specific model to a specific task or domain by learning soft prompts or a lightweight prompt generator attached to the target model. In contrast, GPS is trained once, on a mixture of verifiable reasoning meta-tasks, and then applied as a general-purpose per-sample prompter to downstream tasks for which we do not use any task-specific training data and treat the evaluator as a black box.
>
>   There is also a practical architectural mismatch: IDPG is implemented for encoder-only NLU models (e.g., RoBERTa-style backbones) and sentence-level classification tasks, whereas our experiments use decoder-only generative LLMs on sequence-generation tasks such as summarization and simplification (plus generative classification). Adapting IDPG to this regime would require non-trivial architectural changes and new training objectives, effectively creating a new variant rather than reusing the published method.
>
>
> - *Q5 "Scaling factor for reward terms"*
>
>   We do not use different scaling factors; all reward terms are equally weighted (weight = 1).
>
> - *Q6 "In GRPO, how is the group average advantage calculated?"*
>
>   In our implementation, a group in GRPO is defined as:
>
>   *All rollouts generated by the prompt generator for the same observation-base-prompt pair $(x, b)$ within a single training step.*
>
>   Concretely:
>
>   1. We sample one training example:
>    $$
>    (x, b) \leftarrow \text{SAMPLE}(\mathcal{D}).
>    $$
>   2. The prompt generator produces \(n\) candidate outputs:
>    $$
>    o_{1:n} \sim \pi_{\text{gen}}(\cdot \mid x, b).
>    $$
>   3. From these outputs, we extract $n$ prompts $p_1, \dots, p_n$ and compute their (regularized) scalar rewards: $R_1, \dots, R_n.$
>
>   The set of trajectories (o_j)_{j=1}^{n} and corresponding prompts (p_j)_{j=1}^{n} forms one GRPO group.
>
>   Within this group, we compute the group-average reward:
>   $$ \bar{R} = \frac{1}{n} \sum_{j=1}^n R_j, $$
>   and the group-relative advantage for sample $j$ as:
>   $$A_j = R_j - \bar{R}.$$
>
> - *Q7 "GRPO decoding strategy"*
>
>   We use the standard decoding mechanism with a temperature of 0.9 and top-p of 0.9.

---

### Official Review · Reviewer_gdUi · 2025-11-03

**Soundness:** 2
**Presentation:** 3
**Contribution:** 3
**Rating:** 6
**Confidence:** 3

**Summary:**

This paper introduces GPS, a general-purpose, per-sample prompting framework. GPS generates unique prompts for each input without task-specific fine-tuning. It is trained via RLVR across diverse reasoning and programming tasks, with two regularization techniques (judge and sample regularization) to prevent label leakage. The model demonstrates strong out-of-domain generalization on summarization, simplification, and classification, and on GSM8K.

**Strengths:**

- The problem of per-sample prompting is quite important and well-motivated.
- Method is evaluated across diverse domains—reasoning, summarization, simplification, and classification—with thorough ablations on regularization, decoding (MBR), and per-sample prompting.

**Weaknesses:**

- Given that the prompt generator model is RL-tuned. A fair comparison would be to compare to fully finetune the model on these tasks, with the same utility reward. Would that approach directly outperform the indirect optimization over the prompts? and what would the generalization performance be in that setting?
- I'd want to see a comparison with more recent prompt optimization approaches like in reflective prompt optimization GEPA.

**Questions:**

see above.

---

> ### Author Response · Authors · 2025-11-20
>
> We thank the reviewer for their insightful comments. We address the raised questions below:
>
> - *W1 "Prompt generator model is RL-tuned. A fair comparison would be to compare to fully finetune the model on these tasks.":*
>
>   This is an interesting question. It might be so that finetuning the full model, i.e. including the evaluator model, might bring additional benefits, but doing so would (i) not be transferable to new tasks, as is GPS, which can generate new prompt for tasks unseen in training and (ii) the evaluator has already been fine-tuned on a plethora of other tasks during its post-training (we use the Instruct versions of all involved LLMs), hence understands general instruction following. We argue that fine-tuning the evaluator would also go against the whole idea of prompting, which necessarily freezes the evaluator model.
>
> - *W2: "Comparison against more recent prompt optimization approaches like GEPA":*
>
>   Thank you for the suggestion. We already compare against another recent state of the art prompt generator method, namely PRL. GEPA is primarily evaluated in compound LLM setups with multiple interacting components, rather than the single-evaluator setting we study. We have opted therefore to compare against GRACE, published at NeurIPS 2025, which is directly comparable and targets the same kind of tasks as we do. It is to some degree similar to the ideas used in GEPA, in that it uses a reflection loop to improve upon previously generated prompt candidates. For this comparison please refer to General comments 1. Comparison with GRACE and additional benchmarks.

---

### Author Response · Authors · 2025-11-20
**General comments (4/4)**

References:

[1] N. Lambert et al., “Tülu 3: Pushing frontiers in open language model post-training,” arXiv:2411.15124, 2024.

[2] Muennighoff, Niklas, et al. "s1: Simple test-time scaling." Proceedings of the 2025 Conference on Empirical Methods in Natural Language Processing. 2025.

[3] Lightman, Hunter, et al. "Let's verify step by step." The Twelfth International Conference on Learning Representations. 2023.

[4] Shao, Zhihong, et al. "Deepseekmath: Pushing the limits of mathematical reasoning in open language models." arXiv preprint arXiv:2402.03300 (2024).

[5] Guo, Daya, et al. "Deepseek-r1: Incentivizing reasoning capability in llms via reinforcement learning." arXiv preprint arXiv:2501.12948 (2025).

---

### Author Response · Authors · 2025-11-20
**General comments (3/4)**

**2. Choice of mathematical, coding and commonsense reasoning tasks.**

Regarding the choice of meta-tasks (math, logical reasoning and coding), we intentionally train GPS on verifiable reasoning problems, rather than directly on summarization or classification data, for two reasons. First, these domains naturally provide binary or programmatic rewards (exact numeric answer, multiple-choice correctness, unit tests), which are crucial for stable RL/RLVR-style training. This design is consistent with recent work on “reasoning models,” where reinforcement learning or RL with verifiable rewards on math / logic / coding data leads to broad gains on downstream benchmarks beyond the exact training distribution [1–4]. For example, Tülu 3 [1] applies RLVR on a mixture of verifiable tasks (GSM8K, MATH, and instruction-following data such as IFeval) and reports consistent improvements across diverse benchmarks such as MMLU, GSM8K and HumanEval, even when evaluation tasks differ from the training. Similarly, Muennighoff et al. [2] show in s1: Simple test-time scaling that optimizing on roughly one thousand carefully curated, hard reasoning problems is enough to obtain a model whose test-time–scaled policy transfers across multiple reasoning benchmarks (MATH500, AIME24, GPQA Diamond), outperforming much larger baselines under comparable compute. Work on verifier-based training for math (e.g., Let’s Verify Step by Step) further demonstrates that using verifiable, step-level feedback for multi-step mathematical reasoning significantly improves performance on challenging problems in the MATH benchmark compared to outcome-only supervision [3]. Finally, recent reasoning-focused RL models such as DeepSeekMath and DeepSeek-R1 are trained heavily on math, yet show strong results on a suite of heterogeneous reasoning benchmarks, including competition math, GSM8K, MATH-500, Math Odyssey, LiveCodeBench, and general knowledge benchmarks such as MMLU and GPQA [4,5]. Our setup is in the same spirit: by training GPS on difficult, verifiable reasoning and coding tasks, we encourage it to learn general prompt-construction and reasoning patterns that transfer to unseen NLP tasks (summarization, simplification, classification), even though those tasks never appear during training.

We agree that, in principle, performance will depend on the choice of meta-tasks and reward design; this is also emphasized in recent work on RL with verifiable rewards and process-supervised feedback [1,3]. Exploring larger and more diverse meta-task collections is therefore an interesting extension, but our current results already show that a reasoning-only training mixture can yield non-trivial gains on out-of-domain NLP benchmarks.

---

### Author Response · Authors · 2025-11-20
**General comments (2/4)**

References:

[1] W. Shi, Y. Chen, S. Bian, X. Zhang, K. Tang, P. Hu, Z. Zhao, W. Lu, X. Du. No Loss, No Gain: Gated Refinement and Adaptive Compression for Prompt Optimization. NeurIPS, 2025.

[2] He, Zhiwei, et al. "Deepmath-103k: A large-scale, challenging, decontaminated, and verifiable mathematical dataset for advancing reasoning." arXiv preprint arXiv:2504.11456 (2025).

[3] Lightman, Hunter, et al. "Let's verify step by step." The Twelfth International Conference on Learning Representations. 2023.

[4] Yang, Hang, et al. "Llm-medqa: Enhancing medical question answering through case studies in large language models." arXiv preprint arXiv:2501.05464 (2024).

---

### Author Response · Authors · 2025-11-20
**General comments (1/4)**

**1. Comparison with GRACE and additional benchmarks:**

We thank the reviewers for suggesting comparison with more recent methods and benchmarks. Below, we compare GPS with GRACE [1] (NeurIPS 2025) and extend our evaluation to several additional tasks.

For a fair comparison, we reran GRACE using Qwen2.5-7B-Instruct as both prompt generator and evaluator, instead of the stronger models used in the original paper (DeepSeek-R1 for prompt generation and DeepSeek-V2-0324 for evaluation). We evaluate GRACE and GPS on all benchmarks already included in the paper, as well as on several new benchmarks added specifically for this rebuttal. The GRACE numbers reported here are lower than those in the original GRACE paper, which reflects the difference in underlying model strength rather than the prompt optimization method itself.

We also evaluate GPS on three additional benchmarks:

- MATH500 [3] and DeepMath [2] are large-scale, challenging benchmarks for mathematical reasoning.

  - MATH500 is out-of-distribution for GPS. It was not used for training.

   - DeepMath is in-domain, samples from this dataset were used during training. Training samples do not overlap with test data.

- MedQA [4] is a large-scale, multiple-choice medical QA benchmark based on professional board-exam-style questions. MedQA is also in-domain. We evaluate on the test split not seen during training.

Below we summarize the results.
1. Mathematical Reasoning tasks (DeepMath, Math500, GSM8K):

|MATH500| |
|-|-|

|Method| Accuracy|
|-|-|
|APE|31.53±1.04|
|DE|34.20±1.39|
|GA|40.13±1.39|
|GRACE|33.20±1.60|
|PRL|44.40±1.40|
|**GPS-SR-0.1**|**31.80±1.60**|
|**GPS-J**|**34.20±0.80**|

|DeepMath| |
|-|-|

|Method| Accuracy|
|-|-|
|APE|15.47±0.45|
|GA|18.63±2.37|
|DE|16.10±0.00|
|GRACE|15.05±0.16|
|PRL|21.58±0.22|
|**GPS-SR-0.1**|**20.98±0.08**|
|**GPS-J**|**21.40±0.25**|

DeepMath results above differ slightly from Figure 3 in the paper because we use a smaller maximum generation length (i.e., a lower token budget) for all methods.


|GSM8K||
|-|-|

|Method|Accuracy|
|-|-|
|MI|78.20|
|APE|83.43±1.98|
|GA|81.62±1.38|
|DE|79.52±0.45|
|GRACE| 82.37±1.82|
|PRL| 86.15±0.55|
|**GPS-SR-0.1**|**87.55±0.42**|
|**GPS-J**|**84.45±0.93**|

  2. MedQA (domain base task):

|Method|Accuracy|
|-|-|
|APE|45.66±0.97|
|GA|51.95±1.61|
|DE|51.76±0.16|
|GRACE|52.26±0.16|
|PRL|53.34±0.11|
|**GPS-J**|**54.92±0.14**|
|**GPS-SR-0.1**|**53.31±1.47**|

  3. Classification tasks:

|Method|CR|MR|SST-5|AG's News|SST-2|TREC|Subj|Avg|
|-|-|-|-|-|-|-|-|-|
|MI|87.25|87.40|52.31|82.29|92.70|69.20|57.95|75.59|
|NI|91.50|90.85|51.90|83.43|95.77|66.60|68.10|78.31|
| APO    | 93.48 ± 0.24 | 89.97 ± 1.37 | 53.94 ± 0.29 | 83.73 ± 0.31 | 93.71 ± 0.25 | 71.30 ± 1.90 | 69.80 ± 5.96 | 79.42 |
| APE    | 92.87 ± 0.02 | 89.90 ± 0.94 | 49.37 ± 5.66 | 82.58 ± 1.20 | 91.23 ± 0.66 | 77.07 ± 1.61 | 73.92 ± 1.39 | 79.56 |
| GA     | 92.75 ± 0.40 | 90.45 ± 0.72 | 53.76 ± 1.13 | 82.24 ± 1.00 | 94.65 ± 1.04 | 79.20 ± 2.83 | 74.93 ± 3.12 | 81.14 |
| DE     | 93.38 ± 0.19 | 89.98 ± 0.24 | 55.25 ± 0.37 | 82.18 ± 1.04 | 93.29 ± 0.34 | 76.47 ± 0.38 | 73.08 ± 4.95 | 80.52 |
| GRACE  | 90.92 ± 1.15 | 89.60 ± 1.51 | 53.96 ± 0.93 | 82.34 ± 0.39 | 93.61 ± 0.53 | 72.53 ± 8.62 | 73.92 ± 3.05 | 79.55 |
| PRL    | 92.83 ± 0.24 | 91.27 ± 0.05 | 56.21 ± 0.15 | 84.36 ± 0.08 | 96.32 ± 0.04 | 77.07 ± 2.36 | 76.90 ± 0.95 | 82.14 |
| **GPS-SR-0.1** | **90.50 ± 0.38** | **88.70 ± 0.05** | **55.14 ± 1.13** | **84.21 ± 0.34** | **92.98 ± 0.19** | **68.20 ± 0.20** | **65.10 ± 0.28** | **77.83** |
| **GPS-J** | **90.65 ± 0.05** | **89.15 ± 0.38** | **55.16 ± 0.36** | **84.04 ± 0.02** | **94.25 ± 1.20** | **72.80 ± 0.60** | **64.20 ± 2.25** | **78.61** |


  4. Summarization task:

|Method|ROUGE-1|ROUGE-2|ROUGE-L|
|-|-|-|-|
|MI|32.76|10.39|28.97|
|APE|37.12±2.02|12.97±0.74|33.32±1.68|
|GA|39.69±1.76|14.47±1.00|35.84±1.63|
|DE|33.91±4.04|12.53±1.47|31.05±3.79|
|GRACE|40.61±0.54|14.65±0.53|35.86±0.54|
|PRL|42.47±0.83|16.17±0.24|37.73±0.36|
|**GPS-SR-0.1**|**40.03±0.11**|**14.36±0.13**|**35.91±0.19**|
|**GPS-J**|**38.08±0.74**|**13.07±0.44**|**34.09±0.61**|

  5. Simplification task:

|Method|SARI|
|-|-|
|MI| 43.77|
|APE| 45.33±0.83|
|GA| 46.25±0.47|
|DE| 45.79±0.35|
|GRACE| 50.21±0.18|
|PRL| 52.26±3.51|
|**GPS-SR-0.1**|**52.09±0.22**|
|**GPS-J**|**48.10±0.66**|

We want to stress again that we do not claim that our method will be able to outperform prompt generators that train from scratch on each new task they are tested on, as already evidenced by the comparison against PRL and EvoPrompt. This would be highly surprising, since GPS has only been trained out of distribution. What our work shows is that we can get significant prompt improvements by a general prompt generator that is pre-trained once and then functions for new tasks without any new training. Comparisons against existing prompt generator methods are certainly informative, but are ultimately not comparable in that our method does not need training for new tasks.

---

### Author Response · Authors · 2025-12-02
**Summary Comment**

Dear AC,

We would like to sincerely thank the reviewers and the area chairs for their time and thoughtful feedback. We have carefully considered all comments and addressed the issues raised in our rebuttal, which we believe has further strengthened the paper.

**Strengths**: Reviewers have pointed out the following strengths of our method.

1. Tackles an important and well-motivated problem in general-purpose per-sample prompting without task-specific data (gdUi, hbtD, TVfW, 3EL1),
2. Our judge and sample regularization schemes are novel, reasonable, and effective at preventing reward hacking or label leakage (gdUi, hbtD, TVfW, 3EL1),
3. Our simple two-LLM architecture with a smaller prompter decoupled from the evaluator is practical (TVfW, 3EL1),
4. Broad evaluation and ablations across reasoning, summarization, simplification, and classification, with MBR, per-sample vs task-level prompting, backbone swap, and evaluator scale, and strong GSM8K results are strengths of the paper (gdUi, hbtD, TVfW).

**Weaknesses/Questions**: In our rebuttal, we have addressed the following main points raised by the reviewers.

1. *Additional experiments and strengthened empirical evidence* (gdUi, hbtD, TVfW, 3EL1).

    We introduced a controlled comparison with GRACE (using the same Qwen2.5-7B-Instruct backbone) and new benchmarks: DeepMath and MedQA (in-domain) and MATH500 (out-of-domain), to show that GPS remains competitive on more challenging reasoning tasks and is not underfitting the tasks GPS was trained on.



2. *Clarified positioning vs task-specific methods* (gdUi, hbtD, TVfW, 3EL1).

    We explained in detail how GPS differs from PRL, DSP, IPT/IDPG, and retrieval-based in-context learning. We emphasize that most previous prompt engineering methods (e.g. PRL, GRACE, EvoPrompt-DE, EvoPrompt-GA, APE, APO) are trained separately on each evaluation task with task-specific data, whereas GPS is trained once and then applied zero-shot to unseen tasks without task-specific supervision.

3. *Justified the choice of training tasks and cross-domain generalization* (hbtD, TVfW, 3EL1).

    We clarified why we train on math/logic/coding tasks with verifiable rewards, linked this to recent “reasoning RL” work, and argued that our results demonstrate meaningful transfer from this reasoning-only mixture to unseen NLP tasks such as summarization, simplification, and classification.

4. *Provided technical and regularization details* (hbtD, TVfW, 3EL1).

    We specified the formal learning objective, reward decomposition and scaling, GRPO grouping and advantage computation and decoding settings. We also clarified notation (GPS-J vs GPS-SR-0.1) and how we balance alignment rewards across tasks.

5. *Addressed model-agnosticism and practical concerns* (hbtD, TVfW).

    We showed that retraining GPS with Llama-3.1-8B-Instruct as the generator yields similarly strong (sometimes slightly better) performance, supporting backbone-agnosticity. We reported average prompt lengths (comparable to human prompts and shorter than in-context-example prompts) and discussed the inherent extra LLM call per query as the natural cost of per-sample prompting, in contrast to task-level methods .

We hope this summary helps contextualize our rebuttal.

Best regards,
Authors of submission 474

---

### Meta-Review · Area_Chair_8v5U · 2026-01-06

**Summary:**

The paper introduces GPS, a per-sample prompt generator trained once via RL with verifiable rewards on reasoning/coding tasks and applied zero-shot to unseen tasks. The idea is interesting and well motivated. However, effectiveness remains below task-specific prompt RL (e.g., PRL) on most benchmarks, and the value proposition is not rigorously substantiated with cost-normalized analyses or strong baselines under matched assumptions. Positioning is still weak relative to closely related per-instance methods such as Directional Stimulus Prompting (DSP), instance-dependent prompt tuning, and retrieval-based demonstration selection. Despite added results (DeepMath, MATH500, MedQA) and clarifications (objective, GRPO details, latency/length), the empirical evidence does not convincingly establish superiority or robustness for the claimed general-purpose, no-task-data regime.

**Reviewer Concerns:**

Addressed: The authors clarified the formal objective, reward scaling, GRPO grouping/decoding, variant naming, prompt lengths, and latency trade-offs. They added fairer comparisons to GRACE on a shared backbone and new reasoning benchmarks. They also clarified differences from PRL and discussed backbone-agnosticity.

Outstanding: GPS generally underperforms PRL; the paper lacks a cost-normalized comparison that justifies the claimed amortized advantages. Positioning versus DSP and other per-instance approaches remains insufficiently sharp. Missing baselines under matched assumptions (e.g., demonstration selection when task data are available) limit interpretability. Evaluation breadth still underrepresents harder, non-reasoning tasks and ablations over MBR/best-of-N.

**Reviewer Scores:**

Reviewer Scores

R1 (6): Likely unchanged or slightly down given persisting concerns about effectiveness versus PRL.
R2 (2): May rise to 3–4 due to clarifications, but core empirical concerns remain.
R3 (4): Likely unchanged without stronger positioning vs DSP and stronger baselines.
R4 (4): Likely unchanged; added benchmarks help but do not close the gap.

---

### Decision · Program_Chairs · 2026-01-26

Reject